# A first global height-resolved cloud condensation nuclei data set derived from spaceborne lidar measurements

Goutam Choudhury[1,2] and Matthias Tesche[1]

[1]Leipzig Institute for Meteorology (LIM), Leipzig University, Stephanstrasse 3, 04103 Leipzig, Germany
[2]Department of Geography and Environment, Bar-Ilan University, Ramat Gan 5290002, Israel

**Correspondence:** Goutam Choudhury (goutam.choudhury@biu.ac.il)

**Abstract.** We present a global multiyear height-resolved data set of aerosol-type-specific cloud condensation nuclei concentrations ($n_{CCN}$) estimated from the spaceborne lidar aboard the Cloud-Aerosol Lidar and Infrared Pathfinder Satellite Observation (CALIPSO) satellite. For estimating $n_{CCN}$, we apply the recently introduced Optical Modelling of the CALIPSO Aerosol Microphysics (OMCAM) algorithm to the CALIPSO level 2 aerosol profile product. The estimated $n_{CCN}$ are then gridded into a uniform latitude-longitude grid of $2° \times 5°$, a vertical grid of resolution $60$ m from the surface to an altitude of $8$ km, and a temporal resolution of one month. The data spans a total of 186 months, from June 2006 to December 2021. In addition, we provide a 3D aerosol-type-specific climatology of $n_{CCN}$ produced using the complete time series. We further highlight some potential applications of the data set in the context of aerosol-cloud interactions. The complete data set can be accessed at https://doi.pangaea.de/10.1594/PANGAEA.956215 (Choudhury and Tesche, 2023).

## 1   Introduction

Airborne aerosols can serve as cloud condensation nuclei (CCN) to form liquid cloud droplets and influence the properties of clouds. For instance, an increase in the number of aerosols that can act as CCN may result in more but smaller cloud droplets at a constant cloud liquid water content. As a result, the effective surface area of the cloud available to interact with incoming solar radiation increases, enhancing the cloud albedo and cooling the Earth (Twomey, 1974). Furthermore, it may take longer for these smaller droplets to grow large enough to form precipitation leading to an increase in cloud lifetime and cloud cover, imposing an additional cooling effect (Albrecht, 1989). Understanding and quantifying such aerosol-cloud interactions (ACIs) is necessary for predicting and mitigating their potential impacts on the Earth's radiation budget (Forster et al., 2021).

Only a fraction of aerosols, depending primarily on their size and, to a lesser extent, chemical composition, can act as CCN (Dusek et al., 2006). This dependency is further regulated by atmospheric water vapour supersaturation, which is a function of meteorological parameters like updraft velocity, humidity, temperature, and pressure (Seinfeld et al., 2016). The supersaturation near the cloud base of boundary layer clouds is usually about 0.15-0.20 %. At such conditions, continental and marine aerosols with a dry radius greater than 50 nm ($n_{50,dry}$) and dust aerosols with a dry radius greater than 100 nm ($n_{100,dry}$) form the reservoir of most favourable CCN (Mamouri and Ansmann, 2016; Choudhury and Tesche, 2022b). The concentration of such particles varies with geographical location and as per the findings from surface in-situ measurements, can range from about

$10 \text{ cm}^{-3}$ in pristine conditions to as much as $10^5 \text{ cm}^{-3}$ in polluted urban air masses (Schmale et al., 2018). Similar orders of magnitude variation of CCN concentrations is also seen along the vertical dimension with the highest concentrations close to surface and only a few particles in the upper troposphere (Brock et al., 2021; Zhang et al., 2022). Given the wide range of CCN concentrations, global information on the horizontal and vertical distribution of CCN is necessary to accurately quantify the impact of ACIs on climate (Bellouin et al., 2020; Quaas et al., 2020).

Today, there is a lack of observation-based global CCN data. While climate model outputs can be used to study the global distribution of CCN, they have been identified to underestimate CCN at surface by a normalized mean error of about 40 % to 80 % depending on the geographical location (Fanourgakis et al., 2019). In-situ measurements, despite being performed continuously with a high temporal resolution, are localized to specific land sites and do not cover oceans. Also, such measurements are usually limited to the surface, as it is not feasible to operate airborne measurements over longer periods. Here, satellite

remote sensing, specifically based on spaceborne lidar, emerges as the best way for obtaining a height-resolved, long-term, and observation-based global picture of atmospheric CCN concentrations.

The polar orbiting Cloud-Aerosol Lidar and Infra-Red Pathfinder Satellite Observation (CALIPSO) satellite includes Cloud-Aerosol Lidar with Orthogonal Polarization (CALIOP), which provides height-resolved information on aerosol optical properties for different aerosol types (Winker et al., 2009). Choudhury and Tesche (2022a) present an algorithm that uses the

spaceborne lidar measurements and the aerosol microphysics included in the CALIPSO aerosol model (Omar et al., 2009) to estimate height-resolved cloud-relevant aerosol number concentrations ($n_{50,\text{dry}}$ and $n_{100,\text{dry}}$) and CCN concentrations. The results are found to be consistent with in-situ measurements at various geographic locations over land and ocean covering different aerosol environments (Choudhury et al., 2022; Choudhury and Tesche, 2022b; Aravindhavel et al., 2023). This highlights the potential of CALIOP measurements for producing an unprecedented global CCN data set from satellite observations.

In this paper, we present a global monthly 3D data set of number concentration of CCN ($n_{\text{CCN}}$) for different aerosol types that has been constructed by applying the CCN retrieval algorithm of Choudhury and Tesche (2022a) to more than 15 years (June 2006 to December 2021) of CALIPSO level 2 aerosol profile data. We also provide seasonal and annual climatologies of $n_{\text{CCN}}$ that is estimated from the monthly data. All data sets are produced at a uniform latitude–longitude grid of $2° \times 5°$ resolution and a vertical resolution of $60 \text{ m}$, consistent with the grid of the CALIPSO level 3 monthly product (Tackett et al.,

2018). The data records are given in Network Common Data Form (NetCDF) format separately for each year and can be accessed at https://doi.pangaea.de/10.1594/PANGAEA.956215. Details regarding the input data and algorithm used to produce the CCN data set are discussed in Section 2. Section 3 provides the description of the pre- and post-processing steps involved in the production of the gridded data and elaborates on the physical meaning and application of all the variables stored in the data. The estimated seasonal and annual $n_{\text{CCN}}$ climatologies and a comparison of the annual averaged $n_{\text{CCN}}$ with the global

climate model outputs from Fanourgakis et al. (2019) for the year 2011 are presented in Section 4. We conclude the paper with potential applications of the global CCN data set in Section 5.

## 2 Data and CCN retrieval

### 2.1 CALIPSO aerosol profile product

The CALIPSO polar orbiting satellite has been operational since June 2006. CALIOP, a nadir-viewing elastic backscatter lidar, is the principal payload on CALIPSO. It measures vertical profiles of aerosol and cloud properties at two wavelengths: 532 and 1064 nm. In this work, we use aerosol optical properties at 532 nm from the CALIPSO level 2 version 4.20 (4.21 since July 2020) aerosol profile product (NASA/LARC/SD/ASDC, 2018) such as aerosol extinction coefficient ($\alpha$), backscatter coefficient ($\beta$), and depolarization ratio ($\delta$). All parameters are provided at an along-track horizontal resolution of 5 km with a vertical resolution of 60 m for tropospheric aerosols. We further use the Atmospheric Volume Description flag to access the information on aerosol type, which contains seven categories: marine, desert dust, polluted continental, clean continental, elevated smoke, polluted dust, and dusty marine. These aerosol types are classified based on the type of underlying surface, aerosol optical properties such as the column-integrated attenuated backscatter coefficient and the initial estimated particle depolarization ratio at 532 nm, and the height of the retrieval (Kim et al., 2018). Additionally, quality control flags are utilized to screen the data for the most reliable retrievals (further details given in Section 3). We also use the auxilliary profiles of ambient relative humidity (RH) and pressure in the level 2 data product. These are derived from the Global Modelling and Assimilation Office (GMAO) data assimilation system (GDAS; Molod et al., 2015). The level 2 aerosol profile data considered in this study cover a total of 186 months in the time period from June 2006 to December 2021 (data for February 2016 are not available). Note that CALIPSO underwent an orbit adjustment in September 2018 to synchronize its path with that of the CloudSat satellite. Although this orbital shift resulted in a slight variation in the geographic region observed by CALIPSO, there are currently no known issues associated with CALIPSO's retrieval quality as a result of this transition.

### 2.2 Model data

We use the modelled $n_{CCN}$ from Fanourgakis et al. (2019) for comparison to CALIPSO estimates. The data comprises of annual average surface $n_{CCN}$ at different supersaturations for the year 2011 with a latitude-longitude resolution of $1° \times 1°$ and was obtained from the output of 15 global models. Details on the model configurations and aerosol emission inventories used for modelling $n_{CCN}$ are described comprehensively in Fanourgakis et al. (2019). The data can be downloaded from https://doi.org/10.5281/zenodo.3265866 (last access on June 25, 2023).

### 2.3 CCN retrieval algorithm

The CCN retrieval algorithm used in this work is based on the optical modelling of CALIPSO aerosol microphysics (OMCAM). The algorithm is described comprehensively in Choudhury and Tesche (2022a) and Choudhury et al. (2022). In this section, we briefly review the main steps involved in OMCAM.

OMCAM algorithm first estimates the aerosol size distribution by using: (i) CALIPSO-derived aerosol type-specific extinction coefficients, (ii) their corresponding microphysical properties (normalized volume size distributions and refractive indices),

and (iii) an optical modelling package called Modelled Optical Properties of enseMbles of Aerosol Particles (MOPSMAP; Gasteiger and Wiegner, 2018) for light-scattering calculations. The microphysical properties of continental, dust, and smoke aerosols are taken from CALIPSO's aerosol model (Omar et al., 2009). For marine aerosols, the microphysical properties are derived from Sayer et al. (2012) because it yields aerosol number concentrations that agree better with airborne in-situ measurements (Choudhury et al., 2022). OMCAM first selects a normalized volume size distribution and refractive index based on the aerosol type identified by CALIPSO. The algorithm then scales the normalized size distribution linearly to reproduce the CALIPSO-estimated extinction coefficient as:

$$\frac{dV(r)}{d\ln r} = S_V \cdot \sum_{i=1}^{2} \frac{\nu_i}{\sqrt{2\pi}\ln\sigma_i} \exp\left(\frac{-(\ln r - \ln\mu_i)^2}{2\ln\sigma_i^2}\right) \qquad (1)$$

and $S_V = \dfrac{\alpha}{\alpha_n}$,

where $S_V$ is called the volume scaling factor estimated from the ratio of $\alpha$, the CALIOP-derived extinction coefficient, and $\alpha_n$, the extinction coefficient calculated from the normalized size distribution and refractive index using MOPSMAP. For modelling $\alpha_n$, we treat continental (clean continental, polluted continental, and smoke) and marine aerosols as spheres and use Mie scattering theory. We consider desert dust aerosols as spheroids and use a combination of the T matrix and the improved geometric optics method, depending on the value of aerosol size parameter, to model $\alpha_n$. The values of normalized size distribution parameters such as standard deviation ($\sigma_i$), volume fraction ($\nu_i$), and mean radius ($\mu_i$) of $i^{\text{th}}$ (fine and coarse) mode are listed in the Table A1. After the scaling step, the volume size distribution is converted to a number size distribution, which is then integrated starting at 50 nm to compute $n_{50,\text{dry}}$ for continental and marine aerosols and at 100 nm to compute $n_{100,\text{dry}}$ for desert dust aerosols. This is because aerosols within these size ranges have been identified to act as CCN for a supersaturation of 0.15-0.20 % in several studies through in-situ measurements (Koehler et al., 2009; Rose et al., 2010; Deng et al., 2011; Kumar et al., 2011; Mamouri and Ansmann, 2016). In accordance with the AERONET size distributions, the upper radius limit for integrating the size distributions is set at 15 µm. Further, the CCN concentrations at higher supersaturation can be estimated following the simple CCN parameterization given by Mamouri and Ansmann (2016) as follows:

$$n_{\text{CCN}} = f_{\text{ss}} \cdot n_{j,\text{dry}}, \qquad (2)$$

where $j$ represents the lower radius limit of the size-distribution integration and $f_{\text{ss}}$ is the enhancement factor with values equal to 1.0, 1.35, and 1.7 for supersaturations of 0.15-0.20%, 0.25%, and 0.40%, respectively. While the simple parameterization based on aerosol size and type do not consider the aerosol chemistry stringently, they have been shown to yield reasonable $n_{\text{CCN}}$ estimates, particularly when applied to spaceborne lidar measurements (Choudhury and Tesche, 2022a, b). Further information on the limitations and known issues of the algorithm are discussed in Section 3.3.

## 3    Methodology

In this section, we discuss the complete methodology used to apply the CCN retrieval algorithm to the CALIPSO profile data to generate a global gridded monthly CCN data set. It includes a number of pre-processing stages that are implemented prior to applying the CCN-retrieval algorithm, followed by a set of post-processing steps to convert the $n_{\mathrm{CCN}}$ profiles to global gridded
data.

### 3.1    Pre-processing

#### 3.1.1    Quality screening

The CALIPSO level 2 aerosol profile product includes a number of quality control flags that are intended to filter out unreliable retrievals that may result from clouds misclassified as aerosols and errors pertaining to the extinction coefficient retrieval
(Tackett et al., 2018). The Cloud Aerosol Discrimination (CAD) score specifies the confidence in the classification of aerosol and cloud for each data bin (Liu et al., 2009). We only select data which have a CAD score in the range $[-100, -20]$ as they correspond to a high confidence in the aerosol classification (Liu et al., 2009). To screen retrievals with extinction-coefficient-related retrieval issues, we use the "extinction uncertainty" and the "extinction QC" (extQC) metrics. Data bins with an extinction uncertainty of $-99.99$ km are not considered including the bins present below them as the extinction-
retrieval uncertainty may propagate to solutions at lower altitudes. The extQC flag describes how the extinction coefficient is generated for each level 2 sample. Following Tackett et al. (2018), we only consider bins with extQC values of 0, 1, 16, and 18, because they represent high confidence in the retrieved extinction coefficient. We also use the minimum laser energy (MLE) parameter at 532 nm, which was recently introduced in version 4.20 update, to identify and screen columns affected by low laser energy shots (MLE < 0.08) that results in higher noise and degraded retrieval quality. Additionally, we filter out the
profiles or columns with cloudy pixels because clouds can impede aerosol-retrievals underneath them due to signal attenuation. All the quality screening criteria used in this work are listed in Table 1. Readers are advised to refer to Tackett et al. (2018) for further information on the production and functions of the quality screening flags.

#### 3.1.2    Dust separation

Before applying the CCN retrieval algorithm to the quality screened level 2 data, the extinction coefficients of dust mixtures
(polluted dust and dusty marine) needs to be separated into dust and non-dust components. This is done by using the particle depolarization ratio and first separating the backscatter coefficient into dust and non-dust components following the methodology described in Tesche et al. (2009). We assume the aerosol mixture to be pure dust (non-dust) if the particle depolarization ratio is $> 0.31$ ($< 0.05$). The backscatter coefficients of aerosol mixtures with a depolarization ratio between 0.05 and 0.31 are separated using Eq. (14) of Tesche et al. (2009). This is a rather straightforward technique that assumes aerosols to be externally
mixed. It has been implemented and tested in several studies based on ground-based and spaceborne lidar retrievals (Mamouri and Ansmann, 2015, 2016; Choudhury et al., 2022; Choudhury and Tesche, 2022b). The extinction coefficients of dust and

non-dust aerosol components are then estimated by multiplying the respective separated backscatter coefficients with the lidar ratios corresponding to the aerosol type. Following this methodology, the polluted dust aerosol mixture is separated into polluted continental and desert dust components, and the dusty marine is separated into desert dust and marine components. The lidar ratio for these aerosol types is taken from Kim et al. (2018, Table 2).

### 3.1.3 Hygroscopicity correction

Hydrophilic aerosols may grow in size under moist conditions and thus may result in a higher extinction coefficient relative to dry conditions, even if the aerosol number concentrations remains the same (Zieger et al., 2013). Further, the normalized size distributions used in OMCAM have been derived mostly under dry conditions (Omar et al., 2009) and the CCN parameterization requires dry aerosol number concentration. Therefore, the extinction coefficient is first corrected for aerosol hygroscopicity before applying the OMCAM algorithm. This is done by using the aerosol-type-specific growth factors (ratio of ambient and dry extinction coefficients) given in Choudhury et al. (2022) at different RH. The growth factors for continental and marine aerosols are estimated at different RH by using the kappa paramaterization (Petters and Kreidenweis, 2007), which expresses the hygroscopic growth of an aerosol particle with a dry radius $r_{dry}$ in terms of a hygroscopicity parameter $\kappa$ as

$$\frac{r_{wet}(RH)}{r_{dry}} = (1 + \kappa \cdot \frac{RH}{1 - RH})^{\frac{1}{3}}. \tag{3}$$

Here, $r_{wet}$ is the wet radius of particle at a relative humidity of RH. The ratio between $r_{wet}$ and $r_{dry}$ is used to modify the dry normalized size distributions of aerosols at different RH, followed by the computation of their respective extinction coefficients using MOPSMAP package. A globally averaged kappa value of 0.3 is used for polluted continental, clean continental, and smoke aerosols, and 0.7 is used for marine aerosols (Andreae and Rosenfeld, 2008). We treat desert dust as non-hygroscopic and do not modify the extinction coefficients for dust samples. This approach of has been shown to yield dry extinction coefficients and dry number concentrations that are consistent with in-situ measurements (Choudhury et al., 2022; Choudhury and Tesche, 2022a). Further details on the hygroscopicity correction are given in Choudhury et al. (2022).

## 3.2 Post-processing

We use the quality screened CALIPSO level 2 data obtained by following the steps outlined in Section 3.1.1 in the OMCAM algorithm to estimate $n_{CCN}$ profiles at a supersaturation of 0.20 %. We further limit the $n_{CCN}$ in the output data to a supersaturation of 0.20 % as it can be easily converted to higher supersaturations by using the enhancement factors as shown in Eq. (2). In this section, we discuss the spatial and temporal grid configuration used to covert the $n_{CCN}$ profiles to 3D gridded data. We further describe all the parameters stored in the output data files.

### 3.2.1 Gridding, sampling, and averaging

The CALIPSO level 2 aerosol profile data have a very high vertical resolution of 60 m and an along-swath resolution of 5 km. However, because of the negligible divergence of lasers, CALIPSO has a narrow cross-track coverage. Further, the

distance between two consecutive overpasses is inversely proportional to the latitude because of CALIPSO's polar orbit with an inclination of 98.36°. Therefore, a grid box of a certain dimension close to the equator will include fewer overpasses compared to one close to the poles. Due to this factor and to produce a regionally representative global gridded data with enough aerosol sampling frequency in each grid cell, we choose a monthly temporal resolution and a horizontal grid resolution of $2° \times 5°$ to process the $n_{\mathrm{CCN}}$ profiles. The vertical resolution is, however, kept unchanged at $60\,\mathrm{m}$. We further consider both the daytime and nighttime CALIPSO overpasses to compute the average $n_{\mathrm{CCN}}$ to increase the sampling frequency. This grid configuration is also opted in monthly CALIPSO level 3 products (Tackett et al., 2018) and is suggested by Choudhury and Tesche (2022b) to compile a global $n_{\mathrm{CCN}}$ data set. Using this grid, we estimated the number of days observed with valid $n_{\mathrm{CCN}}$ (value $\geq 0$; hereafter abbreviated as NDO) samples in each grid box for each month and found them to be as high as 17 days in the tropics and 31 days at the poles. Note that the NDO estimated here is different from the "Days_Of_Month_Observed" parameter in CALIPSO level 3 product as the latter also considers valid cloud retrievals.

We further produced a climatological $n_{\mathrm{CCN}}$ data set using the same horizontal and vertical grid configuration. An alternative approach would be to consider a higher horizontal resolution of $1° \times 1°$ as used in Amiridis et al. (2015) to construct a climatology of aerosol and cloud properties using the CALIPSO level 2 data. However, using such a grid results in low aerosol sampling, especially in the tropics. This is shown in Fig. 1 based on more than 15 years of CALIPSO measurements. While the former gives a maximum of 2340 days observed in the tropics, the latter results in a significantly lower maximum of 591 days. Therefore, we use the coarser grid to produce the $n_{\mathrm{CCN}}$ climatology that ensures ample aerosol sampling required for a realistic and regionally representative data set.

For averaging the $n_{\mathrm{CCN}}$ for each latitude, longitude, and altitude grid cell, we follow the methodology used in Tackett et al. (2018) for producing the CALIPSO level 3 aerosol products. We assign the clear-air level 2 bins with an extinction coefficient of $0\,\mathrm{km}^{-1}$, which leads to an $n_{\mathrm{CCN}}$) of $0\,\mathrm{cm}^{-3}$, and compute the average $n_{\mathrm{CCN}}$ ($\overline{n_{\mathrm{CCN}}}$) for each grid cell as:

$$\overline{n_{\mathrm{CCN}}} = \frac{\sum_{i=1}^{N_{\mathrm{a}}} n_{\mathrm{CCN},i}}{N_{\mathrm{a}} + N_{\mathrm{ca}}}, \tag{4}$$

Here, $N_{\mathrm{a}}$ is the number of level 2 samples with aerosol extinction coefficient $> 0\,\mathrm{km}^{-1}$ and $N_{\mathrm{ca}}$ is the number of clear-air samples. This averaging scheme is used in generating the global monthly gridded data set as well as the $n_{\mathrm{CCN}}$ climatology.

### 3.2.2 Output data records

Following the gridding and averaging scheme discussed in the previous section, we produce the global $n_{\mathrm{CCN}}$ data sets at a monthly temporal resolution. Table 2 provides a list of all the parameters included in the output data and their description. Along with the averaged $n_{\mathrm{CCN}}$, we further provide the total number of level 2 samples with aerosol extinction coefficient $\geq 0\,\mathrm{km}^{-1}$ ($N = N_{\mathrm{a}} + N_{\mathrm{ca}}$) and with extinction coefficient $> 0$ ($N_{\mathrm{a}}$). The $n_{\mathrm{CCN}}$, $N$, and $N_a$ are also provided separately for each aerosol type. Note that both $N$ and $N_a$ may not be equal to the sum of the contributions from all the aerosol types, specifically when aerosol mixtures are present. They are useful in computing annual and seasonal averages of $n_{\mathrm{CCN}}$ using Eq. (4). We further provide the NDO for each month and suggest to use the data only when NDO $> 10$ (coverage of a minimum 30 % days of a month). Average pressure and temperature are also provided for each latitude, longitude, and altitude grid cell. The

210 pressure values can be used to convert the data from height coordinates to pressure coordinates. As the retrievals over an altitude of 8 km constitute about 0.7 % of the total tropospheric CCN, to reduce the overall data size, all parameters are limited to a maximum altitude of 8 km. For ease of accessibility across different platforms, we provide the data in Network Common Data Form (NetCDF) format with a medium level of data compression (deflate level of 5). These NetCDF files are accessible at https://doi.pangaea.de/10.1594/PANGAEA.956215 and are tested to work with tools and software like Climate Data Operators

(CDO), netCDF Operators (NCO), and ncdump. The netCDF files with monthly gridded data are provided separately for each year from 2006 to 2021. The $n_{CCN}$ climatology is provided in a separate netCDF file, with data structure and nomenclature similar to the monthly data. The file also includes $n_{CCN}$ climatologies for boreal winter (December, January, and February), spring (March, April, and May), summer (June, July, and August), and autumn (September, October, and November) seasons.

### 3.3 Uncertainty, known issues, and validation

The uncertainty in the estimated $n_{CCN}$ can arise from the uncertainties in the input parameters such as the aerosol extinction coefficient, type-specific normalized size distribution, and the ambient RH as well as from the dust-separation technique and the CCN parameterizations used in the algorithm. Choudhury and Tesche (2022a) studied the sensitivity of the cloud-relevant aerosol number concentrations ($n_{j,dry}$) to the variations in the size distributions for each aerosol type at different RH. They reported a variation of a factor of 1-1.5 depending on the aerosol type. Combining this with the uncertainties associated with

other previously mentioned sources, the overall uncertainty associated with the output $n_{CCN}$ is found to be between a factor of 2 and 3. Such a range is reasonable for spaceborne retrieval of $n_{CCN}$ (Shinozuka et al., 2015; Mamouri and Ansmann, 2016) as atmospheric CCN concentrations can potentially vary by orders of magnitude in space and time (Schmale et al., 2018).

There are some known issues associated with CALIOP's retrieval algorithm and the CCN parameterizations used to produce the global data set. First, faint aerosol layers with extinction coefficient $< 0.001$ km$^{-1}$ (optical depth $< 0.01$) may not exceed

the signal-to-noise ratio required to be detected by CALIOP (Tackett et al., 2018; Mao et al., 2022). The background noise due to solar radiation further impacts the feature detection, especially for the daytime retrievals (Winker et al., 2009, 2013). Such layers may therefore be classified as clear air by CALIOP's feature classification algorithm and assigned with a zero extinction coefficient. This may result in an underestimation of the average extinction and thus the $n_{CCN}$, particularly in grid cells comprising of clean environment (rural continental sites and higher altitudes). Second, the simple aerosol-type-specific

parameterizations implemented in this study assumes all particles over a certain minimum radius to be CCN active. This may lead to an overestimation of $n_{CCN}$ that may also compensate for the underestimation due to the undetected aerosol layers. Third, CALIOP cannot distinguish between polluted continental and smoke aerosol layers that occur below a layer top height of 2.5 km (Kim et al., 2018). Therefore, isolated smoke layers that do not extend above a height of 2.5 km may get classified as polluted continental aerosols, leading to an overestimation of $n_{CCN}$ by about 13.6 %. As a result, we anticipate that the CCN-

retrieval algorithm will overestimate the CCN in areas often influenced by smoke aerosols below 2.5 km. Additionally, strong signal attenuation caused by optically thick aerosol layers located above may lead to increased uncertainties in the retrievals of layers below. Nevertheless, it is anticipated that these retrievals will be filtered out during the quality screening process.

As mentioned earlier in Section 3.2.1, we combine daytime and nighttime CCN retrievals to achieve an optimal sampling frequency. The daytime retrievals usually have a lower signal-to-noise ratio compared to nighttime retrievals, which may result in higher retrieval uncertainty (Young et al., 2013; Tackett et al., 2018). By comparing the daytime and nighttime CCN climatology (refer to Fig. S1 in the supplementary material), we observe higher values over continents in the former. This observation is expected since anthropogenic emissions are more prominent during the day. The values over oceans are comparable in both cases. However, it is important to note that this concurrence may be attributed to long-term averaging used in computing the climatologies. Therefore, a more detailed comparison is required at various temporal scales (instantaneous, monthly, seasonal, and annual) to accurately quantify the effect of merging daytime and nighttime retrievals. Such an investigation is beyond the scope of the present study and will be a subject of future analysis.

The $n_{\mathrm{CCN}}$ estimated using OMCAM algorithm have been validated using independent ground-based and airborne in-situ measurements for different aerosol environments (Choudhury et al., 2022; Choudhury and Tesche, 2022b; Aravindhavel et al., 2023). Even though the uncertainty associated with the CCN-retrieval from CALIOP can be as high as a factor of 3, the validation results indicate a very good agreement with in-situ measurements with a normalized mean bias of $\approx 22$ % and a correlation coefficient of $\approx 0.7$. Such consensus has not yet been achieved for spaceborne estimations of aerosol and CCN number concentrations, accentuating the potential of the spaceborne-lidar derived global $n_{\mathrm{CCN}}$ data set.

## 4  Results

### 4.1  Comparison with global model outputs

A comparison of the CALIOP-estimated average global $n_{\mathrm{CCN}}$ with the model outputs for the year 2011 is shown in Fig. 2. Excluding the data within the Antarctic circle (latitudes $< -66.5°$N), CALIOP-estimated CCN concentrations are, on average, larger than the models, with a normalized difference of ($100 \times (\mathrm{MODEL} - \mathrm{CALIOP})/\mathrm{CALIOP}$) of $-65.6$ %, $89.4$ %, and $4.5$ % for the median, minimum, and maximum of all the models, respectively. The larger values are expected as the models have been identified to underestimate the $n_{\mathrm{CCN}}$ when compared to in-situ measurements at continental surface stations with a mean error of 60 % (Fanourgakis et al., 2019). Best agreement with CALIOP is found for the maximum of all the models, particularly in the Northern Hemisphere. For retrievals over the Southern Ocean with latitudes $< 45°$S, the modelled $n_{\mathrm{CCN}}$ are significantly lower than that of CALIOP with a maximum of $171.9$ cm$^{-3}$. Recent in-situ measurements show that the $n_{\mathrm{CCN}}$ in such regions can even exceed $500$ cm$^{-3}$ (Humphries et al., 2021), which is not observed in the model outputs, though well captured by CALIOP. This highlights the potential of the CALIOP-derived global $n_{\mathrm{CCN}}$ data set in validating model outputs at locations where in-situ observations are sparse or even non-existent. Please note that the comparison presented here is intended to highlight the data set's unique potential for application and that validation studies have already been published (Choudhury et al., 2022; Choudhury and Tesche, 2022b; Fanourgakis et al., 2019). Readers are encouraged to use the full potential of the open-source data set to conduct further comprehensive model evaluation studies.

## 4.2 CCN climatology

Fig. 3 shows the global $n_{CCN}$ climatology computed from the monthly data set averaged for altitudes below 2 km using Eq. 4 and the corresponding averaged $n_{CCN}$ profile. The global average $n_{CCN}$ is estimated to be $365.12 \pm 649.92$ cm$^{-3}$, with $683.25 \pm 1172.95$ cm$^{-3}$ over land and $303.2 \pm 382.33$ cm$^{-3}$ over ocean. Highest concentrations are found over regions influenced by pollution and dust aerosols such as the South and East Asia. Polluted continental aerosols with an average $n_{CCN}$ of $124.3 \pm 459$ cm$^{-3}$ have the maximum contribution to the total CCN below 2 km, followed by marine ($110.8 \pm 175.12$ cm$^{-3}$), dust ($108.9 \pm 256.39$ cm$^{-3}$), and elevated smoke ($21 \pm 74.61$ cm$^{-3}$) aerosols. Clean continental aerosols with a global average of $0.12 \pm 0.34$ cm$^{-3}$ contribute the least to the total CCN (not shown). Over land, the average CCN is estimated to be $683.25 \pm 1172.95$ cm$^{-3}$, with maximum contributions coming from polluted continental ($419.4 \pm 900.9$ cm$^{-3}$), dust ($203.67 \pm 435.45$ cm$^{-3}$), and smoke ($43.6 \pm 117.93$ cm$^{-3}$) aerosols. The concentrations over oceans are less than half of those over land with an average of $303.2 \pm 382.33$ cm$^{-3}$, with marine ($181.93 \pm 221.2$ cm$^{-3}$), dust ($84.68 \pm 206.8$ cm$^{-3}$), and polluted continental ($27.33 \pm 114.17$ cm$^{-3}$) aerosols being the major contributors. Though pollution aerosols contribute the most to the total CCN, it is evident that dust has the most widespread coverage, encompassing nearly the entire globe, indicating its significance in ACI even in pristine aerosol environments far away from the continents.

When considering the vertical distribution of $n_{CCN}$, the highest values are observed near the surface, and these values decrease exponentially as the altitude increases. The majority of marine CCN (97 %) and continental CCN (78 %) are predominantly located at altitudes below 2 km. On the other hand, smoke and dust CCN extend into the free troposphere, with approximately 60 % and 33 % located above 2 km altitude, respectively. Interestingly, smoke CCN exhibit an opposite trend, with concentrations increasing with height and reaching a maximum between 2 and 3 km before decreasing at higher altitudes. Land-based aerosols make up the majority of free tropospheric CCN, with 68 % located above 2 km, compared to 32 % for aerosols over oceans. They exhibit a relatively higher contribution to the global CCN across all altitude levels. It is important to note that this variation may not be observed in localized regions over oceans that are more frequently affected by dust and smoke transported from nearby continents, for instance, the west of Africa.

### 4.2.1 Seasonal climatology

Fig. 4 illustrates the global map presenting the seasonal climatologies of total $n_{CCN}$ for altitudes below 2 km, along with the corresponding profiles of spatially averaged type-specific $n_{CCN}$. The global average $n_{CCN}$ reaches its maximum during winter with a value of $376 \pm 800.91$ cm$^{-3}$. Similar to the global climatology in Fig. 4, the average winter $n_{CCN}$ is significantly higher over land ($725.53 \pm 1499.36$ cm$^{-3}$) compared to over ocean ($321.61 \pm 427.20$ cm$^{-3}$). Strong seasonality in $n_{CCN}$ is observed in regions influenced by Northern Hemisphere summer monsoon, such as Asia and West Africa, with minimum values occurring during the summer. This pattern is likely attributed to the wet scavenging of aerosol particles associated with cloud droplet formation and precipitation during the monsoon season. The seasonality of $n_{CCN}$ is also evident in regions like Eastern and Central Africa, possibly resulting from changes in local biomass burning patterns (Myhre et al., 2003; van der Werf et al., 2017), which peak during dry summer (Southern Hemisphere winter) season.

Furthermore, seasonality in the $n_{\mathrm{CCN}}$ profiles is observed for all the aerosol types, except for marine aerosols. During boreal winter, CCN concentrations are predominantly limited to altitudes below 2-3 km (84-94 %), with the highest near-surface concentrations compared to other seasons. The vertical distribution of CCN for all the aerosol types gradually expands to higher altitudes with the transition to warmer spring and summer seasons. Although near-surface $n_{\mathrm{CCN}}$ are at their lowest during summer, they contribute the most to the free-tropospheric CCN, accounting for 35 % of CCN at altitudes higher than 2 km. This is followed by spring (29 %), autumn (24 %), and winter (16 %), highlighting the substantial impact of boundary layer depth in modulating the vertical extent of CCN throughout all seasons.

It is worthwhile to note that in Fig. 3 and Fig. 4, we have limited our focus to the spatial variations of low-level aerosols (altitude < 2 km) and have only presented the profiles of spatially averaged $n_{\mathrm{CCN}}$. The climatology data includes information up to an altitude of 8 km for various aerosol types, which offers a unique opportunity to investigate the altitudinal variations of type-specific $n_{\mathrm{CCN}}$ across different seasons and regions throughout the globe. Such studies, however, are outside the scope of this paper, which is aimed at introducing a new global CCN data set, and will be the focus of a future publication.

## 5   Conclusion and usage notes

We present a first aerosol type-specific global CCN data set derived from spaceborne lidar measurements at a horizontal latitude-longitude resolution of $2° \times 5°$, a vertical resolution of 60 m, and a temporal resolution of one month. The data set spans more than 15 years, from June 2006 to December 2021, or a total of 186 months. We further use the complete time series to construct a global $n_{\mathrm{CCN}}$ climatology for different aerosol types. The climatologies are also reported at a seasonal time scale. These data sets are aimed at replacing the currently used satellite-derived optical proxies for CCN. Readers are encouraged to utilize the full potential of the global data set for the following purposes:

– The data can be used to investigate the horizontal and vertical distributions of CCN for various aerosol types, as well as to study their trends and variations across monthly, seasonal, and annual timescales. In the Section 4.2 of this paper, we briefly discuss the spatial distribution of $n_{\mathrm{CCN}}$ for different aerosol types for altitudes limited below 2 km, where we find the global average $n_{\mathrm{CCN}}$ to be 365.12 cm$^{-3}$ and identify various CCN hotspots. This approach can be expanded to include the variations along the height and time dimensions. In addition, the type-specific CCN concentrations can be used to identify the sources and sinks of CCN as well as to investigate the contributions of long-range transport of aerosols with high residence time (for example dust and smoke) to CCN in the atmosphere.

– Recent studies have demonstrated that spaceborne aerosol, cloud, and radiation measurements at monthly temporal resolution can be used to estimate the radiative forcing associated with ACIs (Wall et al., 2022; Chen et al., 2022). The monthly data presented in this paper can be similarly coupled with cloud and radiation measurements to quantify ACIs. Studies have demonstrated the effectiveness of CALIOP in even retrieving height-resolved cloud microphysical properties (Zhang et al., 2019; Zang et al., 2021), which can be directly coupled with corresponding CCN estimates to investigate ACIs. Furthermore, the type-specific CCN data can be used to accurately quantify the anthropogenic

component of present-day CCN (sum of polluted continental and smoke components), which was previously estimated from model simulations of past climate (Bellouin et al., 2020; Quaas et al., 2020). This creates a unique opportunity to quantify ACI by solely using satellite observations independent from model simulations.

– Due to the lack of any observation-based global CCN data, model evaluation studies usually rely on in-situ measurements, which are only available for a limited time period at certain point locations. Further, the observations over oceans are sparse and so are the height-resolved measurements restricting the model evaluation to specific land sites. However, validating global model outputs in estimating height-resolved CCN concentrations over both land and ocean is necessary, as they form the key components of ACIs and thus the future climate predictions. With the height-resolved CCN data available for more than 15 years, a comprehensive model evaluation can be performed at different height levels even for regions over oceans far from continents. In Section 4.1 of the paper, we compare the annual average $n_{\mathrm{CCN}}$ estimated from the monthly data with the simulations from 15 global climate models at surface for the year 2011, highlighting the data's usefulness in assessing model outputs. The idea can be further expanded to other models by incorporating the temporal and altitudinal distribution of CCN for different aerosol types encompassing a longer time span.

The above list is by no means exhaustive. Users can also use this data set and check for closure with other spaceborne retrievals of cloud-relevant CCN or aerosol number concentrations, for instance from the polarimetric observations (Hasekamp et al., 2019), the spaceborne lidar using AERONET derived extinction-to-number concentration conversions (Mamouri and Ansmann, 2016), by treating clouds as CCN chambers (Rosenfeld et al., 2016), or any future algorithms or data.

## 6 Code and data availability

The CALIPSO level 2 aerosol profile product can be downloaded from https://doi.org/10.5067/CALIOP/CALIPSO/LID_L2_05KMAPRO-STANDARD-V4-20 (last access: June 25, 2023). The data produced in this work are available at https://doi.pangaea.de/10.1594/PANGAEA.956215 (Choudhury and Tesche, 2023). The model data used for comparison can be found at https://zenodo.org/record/3265866 (last access: June 25, 2023). The codes used to generate the data were written in MATLAB and will be provided by the corresponding author upon reasonable request. The MATLAB live script used to generate the plots shown in the paper is provided in the supplementary files.

*Author contributions.* GC programmed the algorithm, processed the data, prepared the plots, and wrote the initial manuscript. MT contributed to improving the algorithm and the output data set, and to revising the manuscript.

*Competing interests.* The authors declare no conflict of interest.

*Acknowledgements.* We acknowledge the CALIPSO science team for sharing the CALIPSO data. We are grateful to the AERIS/ICARE Data and Services Center for providing access to the CALIPSO data utilised in this work.

*Financial support.* This research has been supported by the Franco-German Fellowship Programme on Climate, Energy, and Earth System Research (Make Our Planet Great Again – German Research Initiative, MOPGA-GRI, grant no. 57429422) of the German Academic Exchange Service (DAAD), funded by the German Ministry of Education and Research.

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

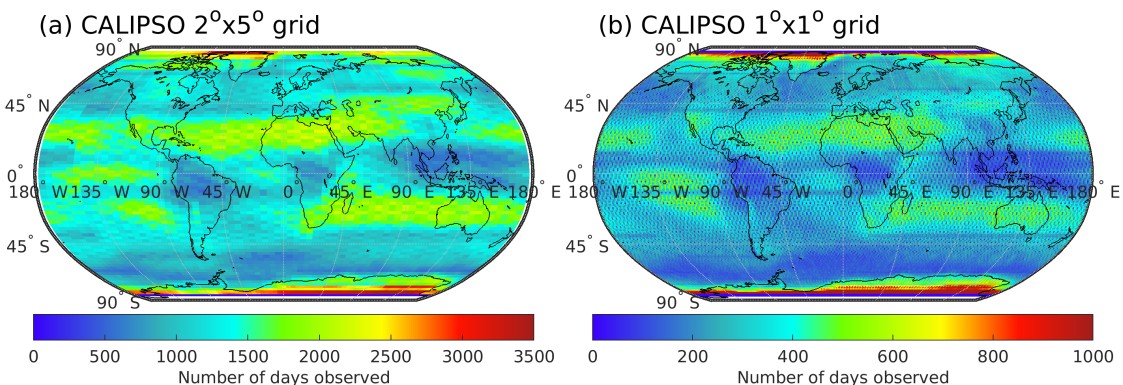

**Figure 1.** Global map of the number of days with valid CCN observations ($n_{\mathrm{CCN}} \geq 0$ cm$^{-3}$) for horizontal grid configurations of $2^\circ \times 5^\circ$ (a) and $1^\circ \times 1^\circ$ (b) produced from 15+ years of CALIOP measurements.

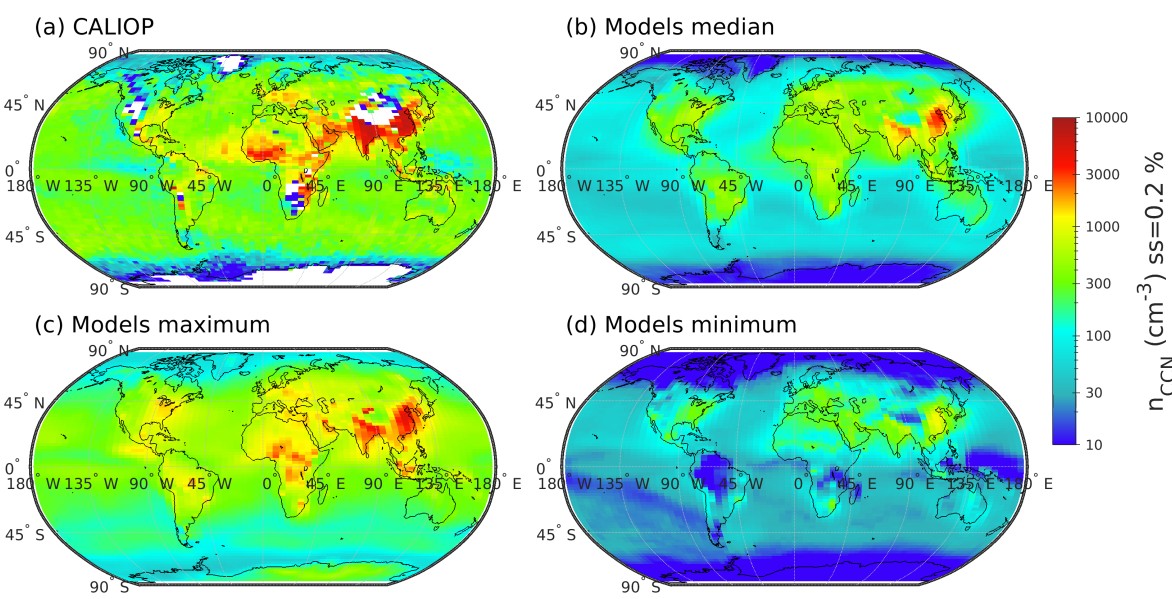

**Figure 2.** Comparison of global $n_{CCN}$ maps at a supersaturation of 0.20 % estimated from CALIOP between altitudes of 500 m and 1 km (a) and from the outputs of 15 global models at surface (b-d) for the year 2011. Panels (b), (c), and (d) represent the median, maximum, and minimum modelled $n_{CCN}$, respectively.

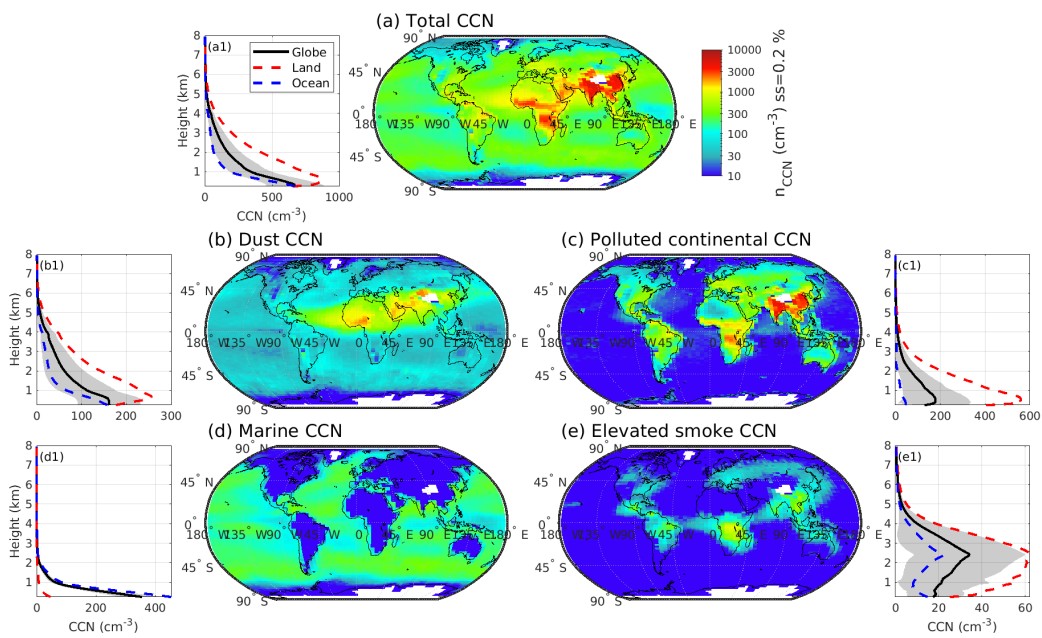

**Figure 3.** Global climatology of $n_{\text{CCN}}$ at a supersaturation of 0.20 % estimated from the monthly CCN data set averaged for altitudes below 2 km for all aerosol types (a), and separated according to mineral dust (b), polluted continental (c), marine (d), and elevated smoke (e) aerosol types, respectively. Sub-panels adjacent to each panel depict the vertical variation in $n_{\text{CCN}}$ averaged over globe (black solid line), land (red dashed line), and ocean (blue dashed line). The semi-transparent grey patch represents half of the standard deviation in globally averaged $n_{\text{CCN}}$.

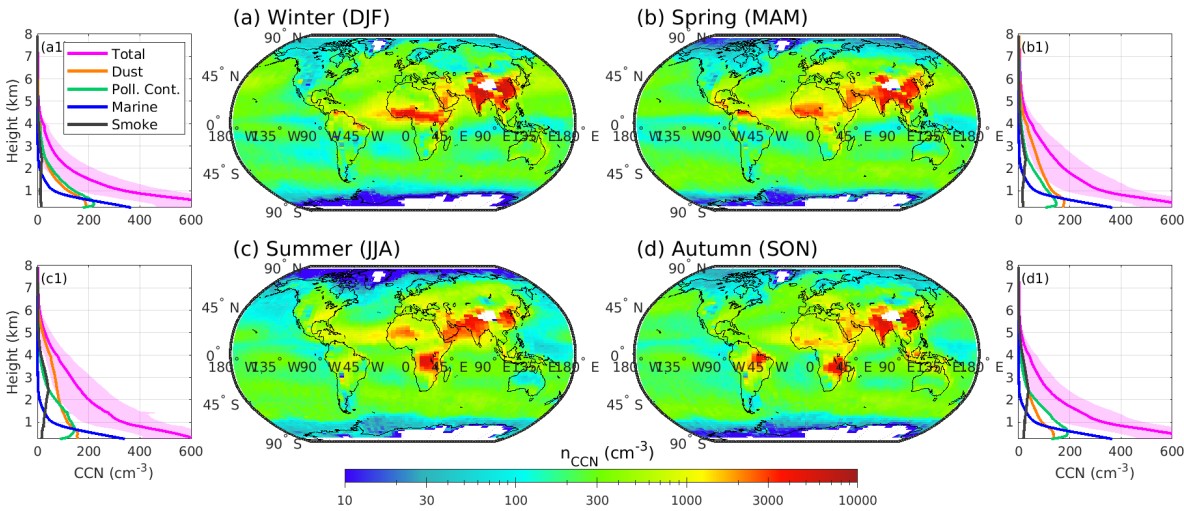

**Figure 4.** Seasonal climatology of $n_{\mathrm{CCN}}$ at a supersaturation of 0.20 % and at altitudes below 2 km for winter (a), spring (b), summer (c) and autumn (d). The vertical variations in seasonal $n_{\mathrm{CCN}}$ is depicted in the adjacent sub-panels for dust (orange line), polluted continental (green line), marine (blue line), and smoke (black line) aerosol types. Magenta profile in the sub-panels represent the total $n_{\mathrm{CCN}}$ and the semi-transparent patch highlights the associated standard deviation scaled to half of its original value for better visualization.

**Table 1.** Quality screening applied to the CALIPSO level 2 aerosol profile product.

| Quality flags | Valid range or values | Comments |
|---|---|---|
| Cloud aerosol discrimination flag | = [-100, -20] | bins outside the bound have low confidence in aerosol classification |
| Extinction QC flag | = 0, 1, 16, and 18 | low confidence in extinction retrievals for other values |
| Extinction uncertainty flag | $\neq$ -99.99 km$^{-1}$ | range bins below the affected one are also rejected |
| Feature type flag 1 | reject if = 2 | reject a profile if it has at least one cloudy bin |
| Feature type flag 2 | reject if = 0 or > 4 | reject any samples if invalid, surface, subsurface, or attenuated |
| Feature type flag 3 | if = 1, set $\alpha = 0$ | extinction coefficient of clear air is assumed to be 0 km$^{-1}$ |
| Minimum laser energy filter | reject if <0.08 | deteriorated data quality in the South Atlantic Anomaly region since mid-2017 |

**Table 2.** Description of the NetCDF data records.

| Parameter | Unit | Description |
|---|---|---|
| lon | °E | longitude mid point |
| lat | °N | latitude mid point |
| altitude | km | altitude mid point above mean sea level |
| **Monthly files** | | |
| time | days | days since 1 January 2000 |
| CCN | $cm^{-3}$ | cloud condensation nuclei (CCN) concentration at 0.2 % ss |
| CCN_std | $cm^{-3}$ | standard deviation of CCN |
| CCN_j<br>j = m, d, pc, cc, es | $cm^{-3}$ | type-specific CCN. m: marine, d: dust, pc: polluted continental, cc: clean continental, es: elevated smoke |
| CCN_j_std | $cm^{-3}$ | standard deviation of CCN_j |
| N | unitless | level 2 bin counts with aerosol extinction coefficient >= 0 |
| N_j | unitless | N for different aerosol types |
| Na | unitless | level 2 bin counts with aerosol extinction coefficient > 0 |
| Na_j | unitless | Na for different aerosol types |
| P | hPa or mb | pressure |
| T | °C | temperature |
| DMO | unitless | number of days observed in a month with N > 0 |
| **Climatology file** | | |
| CCN_cl | $cm^{-3}$ | climatology of CCN |
| CCN_cl_j | $cm^{-3}$ | type-specific CCN climatology |
| CCN_cl_sn | $cm^{-3}$ | seasonal climatology of CCN |
| CCN_cl_sn_j | $cm^{-3}$ | type-specific seasonal CCN climatology |
| NDO | unitless | number of days observed with N > 0 |

**Table A1.** Log-normal bimodal volume size distribution parameters of different aerosol types. $\nu$, $\mu$, and $\sigma$ represent the volume fraction, mode radius, and standard deviation, respectively. Subscript "f" and "c" represent fine anc coarse modes of the size distribution, respectively.

| Aerosol type | Size distribution parameters | | | | | |
|---|---|---|---|---|---|---|
| | $\nu_f$ | $\nu_c$ | $\mu_f$ | $\mu_c$ | $\sigma_f$ | $\sigma_c$ |
| Marine | 0.14 | 0.86 | 0.1137 | 1.8756 | 1.6487 | 2.0544 |
| Dust | 0.223 | 0.777 | 0.1165 | 2.8329 | 1.4813 | 1.9078 |
| Polluted continental | 0.531 | 0.469 | 0.1577 | 3.547 | 1.5257 | 2.065 |
| Clean continental | 0.050 | 0.950 | 0.20556 | 2.6334 | 1.61 | 1.8987 |
| Smoke | 0.329 | 0.671 | 0.1436 | 3.726 | 1.5624 | 2.1426 |