# Peer review of "A first global height-resolved cloud condensation nuclei data set derived from spaceborne lidar measurements"

_Earth System Science Data, 2023_

## Referee Comment (RC1)

**Journal:** Earth System Science Data

**Title:** A first global height-resolved cloud condensation nuclei data set derived from spaceborne lidar measurements

**Manuscript Number:** essd-2023-91

**General Comments:**

This paper releases a long-time series (2006-2021) multilayer dataset of global cloud condensation nuclei concentration (CCN) with the spatial (2°×5°) and temporal (1 month) resolution based on CALIPSO spaceborne lidar. Moreover, the CCN of different aerosol types is also provided based on the developed method by authors. Further, the simple comparison is provided with the results from model. This dataset is important, has large potential to enhance the investigation of aerosol-cloud interaction. The expression and structure of paper is not enough clear now, and should be improved, especially for data process. And the vertical information of estimated CCN should be highlighted, which still missing in this paper. Therefore, I suggest this manuscript can be considered for publication after revision.

**Major Comments:**

1. Introduction. There are some previous works about global retrieval of cloud particle concentration number based on CALIPSO. Author should mention this information. For example, Zhang et al. 2019 (doi.org/10.1364/OE.27.034126) and Zang et al. 2021 (doi.org/10.1364/OE.427022).

2. Title. The retrieval of CCN based on CALIPSO have been tried by previous studies. Please remove "first". It's controversial.

3. Line 66. Please shortly explain how CALIPSO classify the aerosol types to help others understand these aerosol categories clearly.

4. Line 70. The CALIPSO track was change after September 2018. This information should be mentioned in paper. This may be result in some error for the retrieved CCN after September 2018.

5. The length of paragraphs varies so much. For example, Paragraph 1 of section 2.3 is only 3 lines, but following paragraphs are beyond 10 lines. The too long

paragraph possibly let reader miss important information. Please adjust all paragraphs with the proper length throughout this paper.

6. Line 107. What about the upper radius limit of the integration? As many studies already find the CCN and GCCN have even opposite influence on cloud, it's better to indicate the limitation of retrieval for particle size.

7. Line 113. I suggest the writer to add a flow chart containing the steps from Pre-processing to Output data records. The current methodology section is too long and not structured well. This flow chart can be used to clearly introduce the methodology section.

8. Line 130. Authors mention that "we filter out the profiles or columns with cloudy pixels". Do you mean that all profiles containing cloud are removed? Or just remove the vertical bins containing cloud? Additionally, how about the aerosol occurred above cloud?

9. Line 130. CALIPSO signal is gradually attenuated when detecting aerosol and cloud. For the profiles containing multiple aerosol layers, how to consider the effect of attenuation to CCN retrieval, especially for the lower aerosol layer.

10. Line 130. It seems the authors reject all profiles with cloudy pixels. However, the CCN under the cloud is of most interest to scientists. LiDAR can penetrate through some clouds (not too thick) and thus able to obtain aerosol information under the cloud. How about these cloud-penetrated profiles? Do you try it?

11. A main concern (Line 170). The signal-to-noise of CALIPSO have much difference during daytime and nighttime so that aerosol have been retrieved during daytime and nighttime separately (Mao et al. 2022, doi.org/10.5194/acp-22-10589-2022). Do you retrieve the CCN during daytime and nighttime separately? Or consider the uncertainty from this difference.

12. Line 173. How about the number of profiles at the 2°×5°grid for 1 month globally? Such as total profile, cloud-free profile and available aerosol profile. This information (figure) is very useful for reader.

13. Line 200. Does the algorithm have the ability to invert the CCN above 8 km? Can this algorithm be used in smaller time intervals? For example, instantaneous, daily average?

14. Line 208. Does the dataset include uncertainty indicators? At least standard deviation for monthly average data. It is important for others to use.

15. Line 237. Is there a similar pattern to the global distribution of AOD? which can prove the reliability of the CCN product by another way

16. Another main concern. Authors said the retrieved 3D information of CCN based on CALIPSO. This is one of main advantage of the product comparing the resulted CCN based on MODIS. But I do not see that the vertical CCN is highlighted in this paper. It is strange. It is necessary to show the global CCN at different altitude. The profile of CCN also should be shown over several typical regions.

**Minor Comments:**

1. Line 12. It should be revised from "warm clouds" to "clouds".

2. Line 21 and 147. Please add references for these comments.

3. Line 64. Please explain the "(NASA/LARC/SD/ASDC, 2018) ". The reader may be not known CALIPSO well.

4. Line 120. The paragraph is not structured well. Please number all criterions about quality control to let reader can understand easily

5. Line 152. Please add more detailed descriptions about "kappa parameterization".

6. Line 218. The retrieval of instantaneous faint aerosol (or undetected aerosol) has been explored by (Mao et al. 2022, doi.org/10.5194/acp-22-10589-2022) .

7. Please add longitude and latitude labels for all figures.

8. The caption of all figures is not detailed and clear. Such as time range? altitude?

9. Figure 2-4. The color bar should be improved. I suggest the non-value grid is shown as white color, and low-value grid is shown as deep blue color. Because I can't distinguish these white grids are related to low-value or non-value.

10. I do not understand why you copy all figures at supplement information again.

11. Table 1. The comment of extinction QC is not correct

12. Table 2. Adding the information about profile number at each grid into product is better. Such as the number of total profiles, cloud-free profiles and available aerosol profiles. A profile counting available aerosol bins at each grid is also helpful for people who want use this product.

---

## Referee Comment (RC2)

Open Access   Earth System   Discussions
Science
Data

[referee-annotated manuscript omitted]

---

## Author Comment (AC1)

We are thankful to the reviewers for the time, effort, and expertise they have dedicated to reading and reviewing our manuscript. We found their insightful feedback and constructive comments to be helpful in shaping and enhancing the quality of our work. Our point-by-point replies to the comments are provided below. Referee comments are given in black, and our replies are given in blue.

**Reviewer 1**

This paper releases a long-time series (2006-2021) multilayer dataset of global cloud condensation nuclei concentration (CCN) with the spatial (2°×5°) and temporal (1 month) resolution based on CALIPSO spaceborne lidar. Moreover, the CCN of different aerosol types is also provided based on the developed method by authors. Further, the simple comparison is provided with the results from model. This dataset is important, has large potential to enhance the investigation of aerosol-cloud interaction. The expression and structure of the paper is not enough clear now, and should be improved, especially for data process. And the vertical information of estimated CCN should be highlighted, which still missing in this paper. Therefore, I suggest this manuscript can be considered for publication after revision.

Reply: We express our gratitude to the reviewer for dedicating their time to thoroughly read and analyze our work, and for offering valuable comments that have contributed to improving the overall quality and readability of our manuscript. The specific responses to the comments can be found in the subsequent section.

1. Introduction. There are some previous works about global retrieval of cloud particle concentration number based on CALIPSO. Author should mention this information. For example, Zhang et al. 2019 (doi.org/10.1364/OE.27.034126) and Zang et al. 2021 (doi.org/10.1364/OE.427022).

Reply: Thank you for bringing the related articles to our attention. We found them to best suit the discussion in the conclusion section and therefore have added the following to the revised manuscript:

"Studies have demonstrated the effectiveness of CALIOP in even retrieving height-resolved cloud microphysical properties (Zhang et al., 2019; Zang et al., 2021), which can be directly coupled with corresponding CCN estimates to investigate ACIs."

2. Title. The retrieval of CCN based on CALIPSO have been tried by previous studies. Please remove "first". It's controversial.

Reply: Thank you for your suggestion. We agree that there are other CCN retrieval algorithms that exist for application to lidar measurements, either in their development phase or validation phase. However, at the time of submitting the manuscript (and even now), we did not find any global CCN data set based on CALIPSO. We would therefore like to keep the title unmodified.

3. Line 66. Please shortly explain how CALIPSO classify the aerosol types to help others understand these aerosol categories clearly.

Reply: Thank you for your suggestion. We have added the following text to the updated manuscript in Section 2.1.

"These aerosol types are classified based on the type of underlying surface, aerosol optical properties such as the column-integrated attenuated backscatter coefficient and the initial estimated particle depolarization ratio at 532 nm, and the height of the retrieval (Kim et al., 2018)."

4. Line 70. The CALIPSO track was change after September 2018. This information should be mentioned in paper. This may be result in some error for the retrieved CCN after September 2018.

Reply: Thank you for your suggestion. We are aware of the change in CALIPSO's orbit from A-train to C-train following the orbit change of CloudSat. However, there is currently no known issue in the CALIPSO retrieval due to this transition. Please visit CALIPSO's official page on the instrument status for more details, accessible at " https://www-calipso.larc.nasa.gov/tools/instrument_status/inst_status-2018.php " (last accessed on 24 May 2023).

5. The length of paragraphs varies so much. For example, Paragraph 1 of section 2.3 is only 3 lines, but following paragraphs are beyond 10 lines. The too long paragraph possibly let reader miss important information. Please adjust all paragraphs with the proper length throughout this paper.

Reply: Thank you for your suggestion. We understand your point here. However, we design the paragraphs so that each of them conveys a single or very related concept. This results in paragraphs that are sometimes too long or too short. For example, in this case, the first paragraph presents a general overview of the section, so that the readers have an idea of what to expect and where to find further details. The second paragraph discusses the CCN-retrieval algorithm briefly, which has been described comprehensively in our previous articles (Choudhury and Tesche, 2022; Choudhury et al., 2022). Therefore, we have decided to keep the length of the paragraphs unchanged.

6. Line 107. What about the upper radius limit of the integration? As many studies already find the CCN and GCCN have even opposite influence on cloud, it's better to indicate the limitation of retrieval for particle size.

Reply: Thank you for pointing it out. The upper integration limit is 15 μm, which also coincides with that of the AERONET size distribution. We have included this information in the updated manuscript as

"In accordance with the AERONET size distributions, the upper radius limit for integrating the size distributions is set at 15 μm."

7. Line 113. I suggest the writer to add a flow chart containing the steps from Pre-processing to Output data records. The current methodology section is too long and not structured well. This flow chart can be used to clearly introduce the methodology section.

Reply: Thank you for your suggestion. We understand your point here. However, such a flowchart is already given in Choudhury and Tesche (2022) and Aravindhavel et al.

(2023), highlighting the key steps described in the pre-processing and CCN-retrieval algorithm. The only additional information we can add here is the post-processing of the data, which we believe is not significant.

8. Line 130. Authors mention that "we filter out the profiles or columns with cloudy pixels". Do you mean that all profiles containing cloud are removed? Or just remove the vertical bins containing cloud? Additionally, how about the aerosol occurred above cloud?

Reply: Yes, we exclude all profiles containing cloudy pixels, not just the bins. Therefore, profiles with aerosol layers above clouds are also not considered in estimating the CCN data. As per Tackett et al. (2018), this filtering ensures the best quality in level 3 aerosol product. We understand that using this strict filtering may result in a relatively less number of profiles. However, it has been found that CALIPSO underestimates the optical properties of aerosols present above clouds (Kacenelenbogen et al., 2014). For aerosols present below clouds, the retrievals are not trustworthy because of signal attenuation (Tackett et al., 2018). To address the sampling issue resulting from the strict filtering, we have opted for a coarser grid and combined the daytime and nighttime CALIPSO retrievals, as stated in Section 3.2.1 of the manuscript.

9. Line 130. CALIPSO signal is gradually attenuated when detecting aerosol and cloud. For the profiles containing multiple aerosol layers, how to consider the effect of attenuation to CCN retrieval, especially for the lower aerosol layer.

Reply: Usually, a significant level of signal attenuation is caused by clouds. Aerosols, whose concentrations mostly decrease with height, usually do not degrade the signal as much as clouds when present at higher levels. This is the reason why the CALIOP signal usually penetrates the atmosphere, reaching and detecting surfaces even when multiple aerosol layers are present (Tackett et al., 2018). Therefore, we believe that it is not required to account for such an effect in the CCN-retrieval algorithm.

10. Line 130. It seems the authors reject all profiles with cloudy pixels. However, the CCN under the cloud is of most interest to scientists. LiDAR can penetrate through some clouds (not too thick) and thus able to obtain aerosol information under the cloud. How about these cloud-penetrated profiles? Do you try it?

Reply: We understand your concern. However, as stated in the response to Comment 8, the retrievals below clouds have quality issues due to signal attenuation (Tackett et al., 2018) and are hence excluded from the CCN data. Further, the CCN concentrations below clouds may not be the best for studying aerosol-cloud interactions (ACIs), as they are modulated by whether or not the cloud is precipitating (CCN sink). Therefore, we believe that the CCN present in the vicinity of clouds close to the base is the most relevant in the context of ACIs, which are well represented in our data set.

11. A main concern (Line 170). The signal-to-noise of CALIPSO have much difference during daytime and nighttime so that aerosol have been retrieved during daytime and nighttime separately (Mao et al. 2022, doi.org/10.5194/acp-22-10589-2022). Do you retrieve the CCN during daytime and nighttime separately? Or consider the uncertainty from this difference.

Reply: We agree with you. The signal-to-noise ratio (SNR) of CALIOP is comparatively lower for the daytime retrievals, which may lead to faint aerosol layers being undetected (Winker et al., 2009, 2013), as already stated in Line 220 of the unrevised manuscript. We decided to combine the daytime and nighttime retrievals to get a decent sample size sufficient to yield regionally representative CCN data at a monthly resolution. We also believe that it will not impact the mean state of the CCN variations significantly, which is governed by the presence of thick aerosol layers, except in regions that are associated with clean conditions. We have already discussed its consequences in Section 3.3 of the manuscript. The undetected faint aerosol layers during the daytime measurements may lead to an underestimation of CCN concentrations, which may somewhat compensate for the intrinsic overestimation of the CCN-retrieval algorithm that arises from the algorithm approximations. Having said that, we have compiled the daytime and nighttime CCN files separately and are currently working on the comparison, which will be published in a separate study.

12. Line 173. How about the number of profiles at the 2°×5°grid for 1 month globally? Such as total profile, cloud-free profile and available aerosol profile. This information (figure) is very useful for reader.

Reply: Thank you for your suggestion. The data set includes two similar 3D metrics, namely 'Na' (number of aerosol samples) and 'N' (total number of clear air and aerosol samples) detected by CALIPSO within a grid box. Therefore, the maximum value of N in a grid column in a month is equivalent to the total number of profiles used to produce the weighted average CCN. A difference between N and the total number of profiles would arise when the aerosol-related quality flags have impacted many bins within a column. We decided to use and save the N and Na metrics, as they serve a key role in converting the monthly data to seasonal and annual weighted means.

While showing a figure on the number of profiles used in producing the CCN data is important, we believe that the number of days observed (NDO in the manuscript) in a month has better physical meaning in terms of understanding the representativeness of the monthly data set. We already discussed NDO using texts (see Lines 173–176 and 196–198 of the unmodified manuscript) and a figure (see Figure 1) in the manuscript.

13. Line 200. Does the algorithm have the ability to invert the CCN above 8 km? Can this algorithm be used in smaller time intervals? For example, instantaneous, daily average?

Reply: The algorithm is independent of the altitude of the measurements. However, over 8 km, there were not many aerosol bins detected by CALIPSO, with a contribution to the total CCN in terms of magnitude of about 0.7%. Therefore, we excluded such high-altitude retrievals from the data, which would have otherwise increased the data size unnecessarily. We have now included the following in the updated manuscript:

"As the retrievals over an altitude of 8 km constitute about 0.7% of the total tropospheric CCN, to reduce the overall data size, all parameters are limited to a maximum altitude of 8 km."

As stated in Section 3 of the manuscript, the algorithm is designed for application to the CALIPSO level 2 profile product (Choudhury and Tesche, 2022; Choudhury et al., 2022). However, because of the very narrow cross-swath coverage of the lidar (highly collimated beams), the distance between two consecutive CALIPSO overpasses is remarkably high compared to passive sensors like MODIS (Moderate Resolution Imaging Spectroradiometer). In other words, we will not have a CALIPSO overpass in all the grid boxes in a day unless we use a very coarse grid for constructing daily global data. Even going to a monthly resolution with a grid of $2°X5°$ covers a maximum of about 17 days in the tropics, which increases as we go towards the poles. Therefore, using solely CALIPSO observations to produce a daily average would yield unrealistic results. Also, instantaneous global data is not possible as CALIPSO is a polar-orbiting satellite.

14. Line 208. Does the dataset include uncertainty indicators? At least standard deviation for monthly average data. It is important for others to use.

Reply: Unfortunately, we do not include an uncertainty indicator in the data set. However, as stated in Section 3.3 of the manuscript, the uncertainty associated with the CCN concentrations can go as high as a factor of 2 (+100% and -50%). As the CCN concentrations are weighted averages with values ranging many orders of magnitude, we initially planned to include CCN percentiles in the data. However, as the data is 3D and the CCN concentrations are reported for 5 aerosol types, including percentiles nearly tripled the size of the data. Therefore, we decided to omit such a metric.

15. Line 237. Is there a similar pattern to the global distribution of AOD? which can prove the reliability of the CCN product by another way

Reply: Good point. However, we expect the data to be different from AOD, as the latter is an extensive property. Coarser dust aerosols, even though present in small numbers, can have a higher optical depth compared to finer pollution aerosols. Therefore, we compared our data set with the modelled CCN concentrations in the manuscript.

16. Another main concern. Authors said the retrieved 3D information of CCN based on CALIPSO. This is one of main advantage of the product comparing the resulted CCN based on MODIS. But I do not see that the vertical CCN is highlighted in this paper. It is strange. It is necessary to show the global CCN at different altitude. The profile of CCN also should be shown over several typical regions.

Reply: We understand your point. As this is a data descriptor paper, we wanted to focus on explaining the production, usage, and limitations of the data set. As an example of potential application, we compared the data with global model simulations and demonstrated the ability of the data to capture spatial and seasonal CCN variations for aerosols that are relevant to ACI (generally present in the boundary layer below 2 km) in Section 4 of the manuscript. Further studies on the vertical structure of aerosol-type-specific CCN are outside the scope of the current paper but will be discussed comprehensively in future research, as discussed in the usage notes in Section 5 of the manuscript.

**Minor Comments:**

1. Line 12. It should be revised from "warm clouds" to "clouds".

Reply: Corrected.

2. Line 21 and 147. Please add references for these comments.

Reply: Added.

3. Line 64. Please explain the "(NASA/LARC/SD/ASDC, 2018) ". The reader may be not known CALIPSO well.

Reply: It is the standard way of citing CALIPSO level 2 aerosol profile product which is used in this work. Please see "https://asdc.larc.nasa.gov/project/CALIPSO/CAL_LID_L2_05kmAPro-Standard-V4-20_V4-20/citation" (last accessed on 24 May 2023) for more details.

4. Line 120. The paragraph is not structured well. Please number all criterions about quality control to let reader can understand easily

Reply: We apologize for the confusion. However, we have already listed all the quality control criteria in Table 1. We are therefore not modifying the paragraph.

5. Line 152. Please add more detailed descriptions about "kappa parameterization".

Reply: Added. We now have included the parametrization equation and explained how we compute the extinction coefficients at different relative humidities in Section 3.1.3.

6. Line 218. The retrieval of instantaneous faint aerosol (or undetected aerosol) has been explored by (Mao et al. 2022, doi.org/10.5194/acp-22-10589-2022).

Reply: Thank you for your suggestion. We have cited the article in the updated manuscript.

7. Please add longitude and latitude labels for all figures.

Reply: Thank you for your suggestion. We have added the required labels in all the figures.

8. The caption of all figures is not detailed and clear. Such as time range? Altitude?

Reply: Thank you for pointing it out. We missed the altitude range in the Figure 2 caption, which is now added. However, other required information is already given in the figure captions. Figure 1 depicts the NDO (dimension of lat x lon), which does not have an altitude dimension. Figures 3 and 4 captions include the altitude range. From either the caption or the definition of the

plotted parameters (e.g., climatology plots), it is clear that Figures 1, 3, and 4 use the complete data set.

9. Figure 2-4. The color bar should be improved. I suggest the non-value grid is shown as white color, and low-value grid is shown as deep blue color. Because I can't distinguish these white grids are related to low-value or non-value.

Reply: We appologize for the confusion. Thank you for pointing it out. We have modified the color bars used in the figures as per your suggestions.

10. I do not understand why you copy all figures at supplement information again.

Reply: We apologize for the confusion. The supplementary file includes the Matlab live scripts (in pdf format) used to produce the plots from the monthly gridded data presented in the paper. It is intended to help users get started with the data.

11. Table 1. The comment of extinction QC is not correct

Reply: The extinction QC flag depicts the final status of the extinction solution for each aerosol layer (Tackett et al., 2018). Retrievals with extinction QC values of 0, 1, 16, and 18 have the highest confidence as they are associated with no or little change in the initial assumed or computed lidar ratio. The comment in Table 1 for extinction QC in our manuscript says "low confidence in extinction retrievals for other values", which we believe is correct.

12. Table 2. Adding the information about profile number at each grid into product is better. Such as the number of total profiles, cloud-free profiles and available aerosol profiles. A profile counting available aerosol bins at each grid is also helpful for people who want use this product.

Reply: Already addressed in the response to the Comment 12 under the Major comments section.

**Reviewer 2**

The authors describe a data set consisting of global, gridded, height-resolved cloud condensation nuclei (CCN) concentrations retrieved from spaceborne lidar measurements. Such data sets are of importance, e.g., as input and validation data for models, e.g. global circulation models. The authors thoughtfully provide potential applications of their data set and kindly invite readers to use the described data set for further important studies. In general, the manuscript is well written and comprehensible. Some parts are elucidated in more detail, while other parts lack a bit of information which might be of interest to the reader as well (e.g., see comment on dust separation). This data description paper fits well into the scope of Earth Syst. Sci. Data and I suggest publication after minor revisions.

Reply: We are thankful to the reviewer for their thorough examination of our work and for providing valuable constructive feedback that has enhanced the overall quality and readability of our manuscript. We greatly appreciate their time and effort. The detailed responses to their comments can be found in the following section.

1. Line 62: January 1 2021 or July 1 2020? What exactly is meant here? The reference states "The V4.21 data product covers July 1, 2020 to current." (https://asdc.larc.nasa.gov/project/CALIPSO/CAL_LID_L2_05kmAPro-Standard-V4-20_V4-20) Are there significant differences between these two (sub)versions?

Reply: Thank you for pointing this out. You are correct. Version 4.21 product covers from July 1 2020 to present. There are no differences in the scientific algorithms used to generate these two products. The version bump was a result of an upgrade to the operating system used in CALIPSO's data production system.

We have corrected the date in the updated manuscript.

2. Line 71: Why is there no data available for February 2016?

Reply: As per the CALIPSO's official statement, a GPS anomaly resulted in no scientific data collection from 28 January 2016 till 14 March 2016. More information can be found at "https://www-calipso.larc.nasa.gov/tools/instrument_status/inst_status-2016.php" (last accessed on 24 May 2016).

3. Line 129: Add a reference where this Minimum Laser Energy parameter is discribed, please. It is also available for V4.21, I suppose?

Reply: Thank you for pointing it out. We have added the following web reference to the Minimum Laser Energy parameter from CALIPSO's official webpage to the revised manuscript.

"https://www-calipso.larc.nasa.gov/resources/calipso_users_guide/advisory/advisory_2018-06-12.php" (last access: 25 May 2023).

4. Line 144: I suggest to explicitly state the used lidar ratios (Kim et al., 2018) and separation thresholds (particle linear depolarization ratio of dust 0.31? non-dust 0.05?).

Reply: Thank you for your suggestion. We have added the following sentence on the depolarization ratio limits to Section 3.1.2 of the updated manuscript:

"We assume the aerosol mixture to be pure dust (non-dust) if the particle depolarization ratio is > 0.31 (< 0.05). The backscatter coefficients of aerosol mixtures with a depolarization ratio between 0.05 and 0.31 are separated using Eq. (14) of Tesche et al. (2009)."

However, we believe that the lidar ratios can be easily accessed from Kim et al. (2018). Therefore, instead of explicitly stating them, we have indicated that the values are taken from Table 2 of Kim et al. (2018). The updated text in the manuscript is as follows:

"The lidar ratio for these aerosol types is taken from Kim et al. (2018, Table 2)."

5. Line 254ff: Given the provided two digits after comma averages, please also provide standard deviations.

Reply: Thank you for your suggestion. We have now mentioned the standard deviations along with the mean CCN concentrations in the revised manuscript.

6. Similar to Referee 1, I wonder about CCN retrievals at heights >8 km AGL, and even more so, about potential application of this technique (or a similar, adapted one) on INP (ice nucleation particle) retrievals, even though, I understand that a substantial discussion of that would be beyond the scope of a data description paper.

Reply: The algorithm is independent of the altitude of the measurements. However, over 8 km, there were not many aerosol bins detected by CALIPSO, with their contribution to the total CCN in terms of magnitude being about 0.7%. Therefore, we excluded such high-altitude retrievals from the data, which would have otherwise increased the data size unnecessarily. We have now included the following in the updated manuscript:

"As the retrievals over an altitude of 8 km constitute about 0.7% of the total tropospheric CCN, to reduce the overall data size, all parameters are limited to a maximum altitude of 8 km."

Regarding the potential application of the algorithm to retrieve INP from CALIOP, we are still in the development and validation phases. However, another popular technique given by Mamouri and Ansmann (2015, 2016) has shown promising results in retrieving INP concentrations from ground-based and spaceborne lidar measurements (Marinou et al., 2019). As our paper specifically focuses on the CCN data set, we have not included this discussion in our manuscript.

7. Further, rather minor textual recommendations can be found in the attached pdf file.

Reply: Thank you for your recommendations. We have corrected all of them in the revised manuscript.

**References**:

Aravindhavel, A., Choudhury, G., Prabhakaran, T., Murugavel, P., and Tesche, M.: Retrieval and validation of cloud condensation nuclei from satellite and airborne measurements over the Indian Monsoon region, Atmospheric Research, 290, 106 802, https://doi.org/https://doi.org/10.1016/j.atmosres.2023.106802, 2023.

Choudhury, G. and Tesche, M.: Estimating cloud condensation nuclei concentrations from CALIPSO lidar measurements, Atmospheric Measurement Techniques, 15, 639–654, https://doi.org/10.5194/amt-15-639-2022, 2022.

Choudhury, G., Ansmann, A., and Tesche, M.: Evaluation of aerosol number concentrations from CALIPSO with ATom airborne in situ measurements, Atmospheric Chemistry and Physics, 22, 7143–7161, https://doi.org/10.5194/acp-22-7143-2022, 2022.

Kim, M.-H., Omar, A. H., Tackett, J. L., Vaughan, M. A., Winker, D. M., Trepte, C. R., Hu, Y., Liu, Z., Poole, L. R., Pitts, M. C., Kar, J., and Magill, B. E.: The CALIPSO version 4 automated aerosol classification and lidar ratio selection algorithm, Atmospheric Measurement Techniques, 11, 6107–6135, https://doi.org/10.5194/amt-11-6107-2018, 2018.

Mamouri, R. E. and Ansmann, A.: Estimated desert-dust ice nuclei profiles from polarization lidar: methodology and case studies, Atmo- spheric Chemistry and Physics, 15, 3463–3477, https://doi.org/10.5194/acp-15-3463-2015, 2015.

Mamouri, R.-E. and Ansmann, A.: Potential of polarization lidar to provide profiles of CCN- and INP-relevant aerosol parameters, Atmo-395 spheric Chemistry and Physics, 16, 5905–5931, https://doi.org/10.5194/acp-16-5905-2016, 2016.

Marinou, E., Tesche, M., Nenes, A., Ansmann, A., Schrod, J., Mamali, D., Tsekeri, A., Pikridas, M., Baars, H., Engelmann, R., Voudouri, K.-A., Solomos, S., Sciare, J., Groß, S., Ewald, F., and Amiridis, V.: Retrieval of ice-nucleating particle concentrations from lidar observations and comparison with UAV in situ measurements, Atmos. Chem. Phys., 19, 11315–11342, https://doi.org/10.5194/acp-19-11315-2019, 2019.

Petters, M. D. and Kreidenweis, S. M.: A single parameter representation of hygroscopic growth and cloud condensation nucleus activity, Atmospheric Chemistry and Physics, 7, 1961–1971, https://doi.org/10.5194/acp-7-1961-2007, 2007.

Tackett, J. L., Winker, D. M., Getzewich, B. J., Vaughan, M. A., Young, S. A., and Kar, J.: CALIPSO lidar level 3 aerosol profile product: version 3 algorithm design, Atmospheric Measurement Techniques, 11, 4129–4152, https://doi.org/10.5194/amt-11-4129-2018, 2018.

Tesche, M., Ansmann, A., Müller, D., Althausen, D., Engelmann, R., Freudenthaler, V., and Groß, S.: Vertically resolved separation of dust and smoke over Cape Verde using multiwavelength Raman and polarization lidars during Saharan Mineral Dust Experiment 2008, Journal of Geophysical Research: Atmospheres, 114, https://doi.org/10.1029/2009JD011862, 2009.

Kacenelenbogen, M., Redemann, J., Vaughan, M. A., Omar, A. H., Russell, P. B., Burton, S., Rogers, R. R., Ferrare, R. A., and Hostetler, C. A. (2014), An evaluation of CALIOP/CALIPSO's aerosol-above-cloud detection and retrieval capability over North America, *J. Geophys. Res. Atmos.*, 119, 230– 244, doi:10.1002/2013JD020178.

Winker, D. M., Vaughan, M. A., Omar, A., Hu, Y., Powell, K. A., Liu, Z., Hunt, W. H., and Young, S. A.: Overview of the CALIPSO Mission and CALIOP Data Processing Algorithms, Journal of Atmospheric and Oceanic Technology, 26, 2310–2323, https://doi.org/10.1175/2009JTECHA1281.1, 2009.

Winker, D. M., Tackett, J. L., Getzewich, B. J., Liu, Z., Vaughan, M. A., and Rogers, R. R.: The global 3-D distribution of tropospheric aerosols as characterized by CALIOP, Atmospheric Chemistry and Physics, 13, 3345–3361, https://doi.org/10.5194/acp-13-3345-2013, 2013.

Zieger, P., Fierz-Schmidhauser, R., Weingartner, E., and Baltensperger, U.: Effects of relative humidity on aerosol light scattering: results from different European sites, Atmospheric Chemistry and Physics, 13, 10 609–10 631, https://doi.org/10.5194/acp-13-10609-2013, 2013

---

## Author Response (AR2)

We are again thankful to the editor and the reviewer for their time and effort they have dedicated to reviewing our manuscript. We agree with all the points that the reviewer has raised and have incorporated them in the revised manuscript. Our point-by-point replies to the comments are provided below. Referee comments are given in black and our replies are given in blue.

**General Comments:**

Thank you for authors's revision. Unfortunately, the authors response all question, but little resolved the problems I raised. Especially some important results are still missing, e.g. vertical distribution of CCN, uncertainty of dataset and day-night difference. I do not expect authors resolve all problems about lidar retrieval of CCN. But lidar is very sensitive to aerosol particle and enviromental conditions, the nessacery and enough diccusion about the limitationes and undertainties must been provided here. It is responsibility to consider or discuss all possible uncertainties if authors want publish a available dataset. Therefore, I suggest this manuscript should be further revised.

Reply: We appologize for the inconvenience. We have carefully incorporated all your suggestions in the revised manuscript and have revised the CCN data. We provide further details in the specific responses below.

**Major Comments:**

1. For previous Question 4: The CALIPSO track was change after September 2018. At least there is a difference in the observed area, which should be pointed out in the text.

Reply: We understand your point. We have now discussed it in Section 2.1 of the revised manuscript by adding the following sentences to lines 73-75 of the updated manuscript.

"Note that CALIPSO underwent an orbit adjustment in September 2018 to synchronize its path with that of the CloudSat satellite. Although this orbital shift resulted in a slight variation in the geographic region observed by CALIPSO, there are currently no known issues associated with CALIPSO's retrieval quality as a result of this transition."

2. For previous Question 9: For multiple layers of aerosols, the attenuation caused by the upper aerosol layer must influence the retrieval of CCN at lower layer. The author explains the LiDAR can penetrate most of aerosol layers, but how about heavy dust condition, which should be pointed out in the text? I know the CALIPSO data have been calibrated based on signal attenuation, but the related question is still existed. The effect of attenuation to CCN retrieval should been reminded in uncertainties discussion.

Reply: We agree with you. There can be instances where strong signal attenuation due to optically thick aerosol layer present above may lead to increased uncertainty in the retrieval below. Having said that, we do believe that a majority of these uncertain retrievals may be identified and filtered by the several quality control metrics included in the CALIPSO level 2 data, especially the extinction uncertainty metric. We have now discussed the uncertainty that may arise because of this issue in the updated manuscript by adding the following in lines 240-242 of the revised manuscript.

"Additionally, strong signal attenuation caused by optically thick aerosol layers located above may lead to increased uncertainties in the retrievals of layers below. Nevertheless, it is anticipated that these retrievals will be filtered out during the quality screening process."

3. For previous Question 11: I am very surprise author say "We also believe that it will not impact themean state of the CCN variations significantly". The signal-to-noise of CALIPSO during daytime and nighttime even have the difference at one magnitude. The consideration about sample size is not reason why author not consider the difference of retrieval between daytime and nighttime. Addtionally, the day and night CCN retrieval uncertainty should be different due to the different signal-to-noise of CALIPSO. At least, authers should provide a figure and dissusion about CCN retrieval during daytime and nighttime, separately. This is important message for readers to use this dataset, which should be pointed out in the text.

Reply: We understand your point. We appologize for being not clear enough in our previous response. The low signal-to-noise ratio (SNR) of CALIPSO during daytime impacts on the aerosol extinction retrieval and therefore the CCN retrieval (Tackett et al., 2018; Mao et al., 2022). This may lead to weaker detection of aerosol features, especially the faint ones, which may be incorrectly classified as "clear air" by CALIPSO leading to an extinction coefficient (and CCN) underestimation. We tried to convey this information in Section 3.3 of the unmodified manuscript in lines 227-232 using the following:

"First, faint aerosol layers with extinction coefficient < 0.001 $km^{-1}$ (optical depth < 0.01) may not exceed the signal-to-noise ratio required to be detected by CALIOP (Tackett et al., 2018; Mao et al., 2022). The background noise due to solar radiation further impacts the feature detection, especially for the daytime retrievals (Winker et al., 2009, 2013). Such layers may therefore be classified as clear air by CALIOP's feature classification algorithm and assigned with a zero extinction coefficient. This may result in an underestimation of the average extinction and thus the $n_{CCN}$, particularly in grid cells comprising of clean environment (rural continental sites and higher altitudes)."

We agree that the low SNR may also lead to an increased uncertainty in the retrieval. A part of this effect can be observed in the increase in extinction uncertainty metric given in level 2 CALIPSO aerosol profile data (Young et al., 2013). To constrain this limitation, we use the uncertainty metric (including other metrics like CAD score aimed at limiting the highly uncertain retrievals) in our data pre-processing stage and filter the CALIPSO profiles accordingly. The CCN climatology (similar to Figure 3 in manuscript) produced using daytime and nighttime CALIPSO retrievals separately is shown in Figure R1 of this document. As seen in the figure, the daytime retrievals show a higher average CCN value compare to night. This is expected over land as the anthropogenic activities are more pronounced during day light hours. This information can also be derived from the second row of Figure R1 in which the CCN concentrations from polluted continental aerosols have much higher magnitude during the day.

We do agree with the reviewer that the daytime retrievals are more susceptible to retrieval-related errors because of their relatively low SNR ratio and may occassionally result in significantly different "instantaneous" aerosol properties. However, if such occassional uncertain retrieval occur, they are expected to be suppressed over the period of a month (resolution used in our study) by the more frequent error free retrievals. This is what we wanted to convey in the previous response when we mentioned that the mean state of the CCN may not be affected significantly. Nevertheless, we believe that a detailed study is needed to better quantify the difference between daytime and nighttime aerosol properties at different time scales (instantaneous, month, season, and annual). We discuss this topic in Section 3.3 of the manuscript and keep the Figure R1 in the supplementary of the manuscript. Following texts are added to the revised manuscript (lines 243-251).

"As mentioned earlier in Section 3.2.1, we combine daytime and nighttime CCN retrievals to achieve an optimal sampling frequency. The daytime retrievals usually have a lower signal-to-noise ratio (SNR) compared to nighttime retrievals, which may result in higher retrieval uncertainty (Young et al., 2013; Tackett et al., 2018). By comparing the daytime and nighttime CCN climatology (refer to Fig. S1 in the supplementary material), we observe higher values over continents in the former. This observation is expected since anthropogenic emissions are more prominent during the day. The values over oceans are comparable in both cases. However, it is important to note that this concurrence may be attributed to long-term averaging used in computing the climatologies. Therefore, a more detailed comparison is required at various temporal scales (instantaneous, monthly, seasonal, and annual) to accurately quantify the effect of merging daytime and nighttime retrievals. Such an investigation is beyond the scope of the present study and will be a subject of future analysis."

[Figure]

*Figure R1: CCN daytime (left column) and nighttime (right column) climatology estimated using more than 15 years of CALIPSO level 2 aerosol profile product (June 2006 to December 2021). The top and bottom row represents the total CCN and polluted continental CCN concentrations, respectively.*

4. For previous Question 14: Line 208. The uncertainty indicators are so important. For monthly average data, the authors do not need to provide many percentiles data, but at least the standard deviation for each pixel also can provide some information for using data.

Reply: We agree with you. We have now revised the monthly CCN data by adding standard deviations for mean total CCN concentrations and mean aerosol type-specific CCN concentrations. We have updated Table 2 of the manuscript accordingly. The

following variables are added to the updated data set: CCN_std, CCN_cc_std, CCN_pc_std, CCN_d_std, CCN_m_std, and CCN_es_std.

5. For previous Question 16: If the authors emphasize height resolved or 3D, it is necessary to provide a profile of CCN distribution or global CCN at some typical altitude levels. I do not understand authors annoucement the dataset is vertical, but not provide the vertical information. Maybe, authors remove the "height-resolved" from title.

Reply: We apologize for not presenting the vertical profiles of CCN concentrations. We have now updated Figures 3 and 4 of the manuscript by adding an additional sub-panel to each panel showing the CCN profiles. In the CCN climatology shown in Figure 3, we now include the global mean CCN profile and also separate profiles for CCN over land and ocean for each aerosol type. Further, in the CCN seasonal climatology (Figure 4), we show the CCN profile for different aerosol type for four (boreal) seasons. We further discuss the CCN profiles in Section 4.2 of the manuscript. The following texts are added to the revised manuscript.

"When considering the vertical distribution of $n_{CCN}$, the highest values are observed near the surface, and these values decrease exponentially as the altitude increases. The majority of marine CCN (97 %) and continental CCN (78 %) are predominantly located at altitudes below 2 km. On the other hand, smoke and dust CCN extend into the free troposphere, with approximately 60 % and 33 % located above 2 km altitude, respectively. Interestingly, smoke CCN exhibit an opposite trend, with concentrations increasing with height and reaching a maximum between 2 and 3 km before decreasing at higher altitudes. Land-based aerosols make up the majority of free tropospheric CCN, with 68 % located above 2 km, compared to 32 % for aerosols over oceans. They exhibit a relatively higher contribution to the global CCN across all altitude levels. It is important to note that this variation may not be observed in localized regions over oceans that are more frequently affected by dust and smoke transported from nearby continents, for instance in regions close to the west coast of African continents."

"Furthermore, seasonality in the $n_{CCN}$ profiles is observed for all the aerosol types, except for marine aerosols. During boreal winter, CCN concentrations are predominantly limited to altitudes below 2-3 km (84-94 %), with the highest near-surface concentrations compared to other seasons. The vertical distribution of CCN for all the aerosol types gradually expands to higher altitudes with the transition to warmer spring and summer seasons. Although near-surface $n_{CCN}$ are at their lowest during summer, they contribute the most to the free-tropospheric CCN, accounting for 35 % of CCN at altitudes higher than 2 km. This is followed by spring (29 %), autumn (24 %), and winter (16 %), highlighting the substantial impact of boundary layer depth in modulating the vertical extent of CCN throughout all seasons."

**Other modifications:**

As mentioned in the response to comment 3, we have now added Figure R1 to the supplementary (as Fig. S1). We have further updated the MATLAB scripts in the supplementary.

**References:**

Tackett, J. L., Winker, D. M., Getzewich, B. J., Vaughan, M. A., Young, S. A., and Kar, J.: CALIPSO lidar level 3 aerosol profile product: version 3 algorithm design, Atmos. Meas. Tech., 11, 4129–4152, https://doi.org/10.5194/amt-11-4129-2018, 2018.

Winker, D. M., M. A. Vaughan, A. Omar, Y. Hu, K. A. Powell, Z. Liu, W. H. Hunt, and S. A. Young, 2009: Overview of the CALIPSO Mission and CALIOP Data Processing Algorithms. J. Atmos. Oceanic Technol., 26, 2310–2323, https://doi.org/10.1175/2009JTECHA1281.1.

Winker, D. M., Tackett, J. L., Getzewich, B. J., Liu, Z., Vaughan, M. A., and Rogers, R. R.: The global 3-D distribution of tropospheric aerosols as characterized by CALIOP, Atmos. Chem. Phys., 13, 3345–3361, https://doi.org/10.5194/acp-13-3345-2013, 2013.

Young, S. A., Vaughan, M. A., Kuehn, R. E., and Winker, D. M.: The Retrieval of Profiles of Particulate Extinction from Cloud–Aerosol Lidar and Infrared Pathfinder Satellite Observations (CALIPSO) Data: Uncertainty and Error Sensitivity Analyses, J. Atmos. Ocean. Tech., 30, 395–428, https://doi.org/10.1175/jtech-d-12-00046.1, 2013.